# Association of cardiometabolic risk factors with hospitalisation or death due to COVID-19: population-based cohort study in Sweden (SCAPIS)

Per Tornhammar,[1] Tomas Jernberg,[2] Göran Bergström,[3,4] Anders Blomberg,[5] Gunnar Engström,[6] Jan Engvall,[7,8] Tove Fall [ID],[9] Magnus Gisslén,[10,11] Christer Janson [ID],[12] Lars Lind,[13] C Magnus Sköld,[14,15] Johan Sundström,[13,16] Stefan Söderberg [ID],[5] Suneela Zaigham,[6] Carl Johan Östgren,[8] Daniel Peter Andersson [ID],[17] Peter Ueda [ID][18]

DPA and PU are joint senior authors.

For numbered affiliations see end of article.

**Correspondence to**
Dr Peter Ueda; peter.ueda@ki.se

## ABSTRACT

**Objective** To assess the association of cardiometabolic risk factors with hospitalisation or death due to COVID-19 in the general population.

**Design, setting and participants** Swedish population-based cohort including 29 955 participants.

**Exposures** Cardiometabolic risk factors assessed between 2014 and 2018.

**Main outcome measures** Hospitalisation or death due to COVID-19, as registered in nationwide registers from 31 January 2020 through 12 September 2020. Associations of cardiometabolic risk factors with the outcome were assessed using logistic regression adjusted for age, sex, birthplace and education.

**Results** Mean (SD) age was 61.2 (4.5) and 51.5% were women. 69 participants experienced hospitalisation or death due to COVID-19. Examples of statistically significant associations between baseline factors and subsequent hospitalisation or death due to COVID-19 included overweight (adjusted OR (aOR) vs normal weight 2.73 (95% CI 1.25 to 5.94)), obesity (aOR vs normal weight 4.09 (95% CI 1.82 to 9.18)), pre-diabetes (aOR vs normoglycaemia 2.56 (95% CI 1.44 to 4.55)), diabetes (aOR vs normoglycaemia 3.96 (95% CI 2.13 to 7.36)), sedentary time (aOR per hour/day increase 1.10 (95% CI 1.02 to 1.17)), grade 2 hypertension (aOR vs normotension 2.44 (95% CI 1.10 to 5.44)) and high density lipoprotein cholesterol (aOR per mmol/L increase 0.33 (95% CI 0.17 to 0.65)). Statistically significant associations were not observed for grade 1 hypertension (aOR vs normotension 1.03 (95% CI 0.55 to 1.96)), current smoking (aOR 0.56 (95% CI 0.24 to 1.30)), total cholesterol (aOR per mmol/L increase 0.90 (95% CI 0.71 to 1.13)), low density lipoprotein cholesterol (aOR per mmol/L increase 0.90 (95% CI 0.69 to 1.15)) and coronary artery calcium score (aOR per 10 units increase 1.00 (95% CI 0.99 to 1.01)).

**Conclusions** In a large population-based sample from the general population, several cardiometabolic risk factors were associated with hospitalisation or death due to COVID-19.

## Strengths and limitations of this study

► This study used data on cardiometabolic risk factors measured between 2014 and 2018 in a population-based cohort of almost 30 000 participants and assessed their association with hospitalisation or death due to COVID-19 during the first wave of the pandemic.

► Few previous studies have used population-based samples from the general population to assess the relationship between cardiometabolic risk factors and outcomes in COVID-19.

► As we could not capture all cases of COVID-19 which did not lead to hospitalisation or death, we could not assess to what extent the observed associations may reflect the relationship with exposure to SARS-CoV-2 as compared with the risk of hospitalisation or death due to COVID-19 among those who have been exposed to the virus.

Identification of individuals at risk of worse outcomes in COVID-19 may inform risk management decisions to mitigate exposure and to prioritise vaccination. Several studies have shown that cardiometabolic risk factors are associated with a higher risk of adverse outcomes in COVID-19. While many of these studies have been performed in selected populations such as patients hospitalised with COVID-19[1 2] or patients with certain diagnoses,[3 4] fewer studies have been based on data from the general population.[5–8]

The Swedish CArdioPulmonary bioImage Study (SCAPIS)[9] is a population-based cohort of approximately 30 000 men and women who were extensively characterised with respect to cardiometabolic risk factors and function at the age of 50–64 years during the years preceding the COVID-19 outbreak

(2014 through 2018). We used data from SCAPIS to assess the association of cardiometabolic risk factors with risk of hospitalisation or death due to COVID-19 during the first wave of the pandemic.

## METHODS
### Data sources
The SCAPIS cohort, described in detail elsewhere,[9] is a population-based cohort conducted at six Swedish university hospitals in Gothenburg, Linköping, Malmö/Lund, Stockholm, Umeå and Uppsala, with each site recruiting participants from corresponding municipality areas. From 2014 through 2018, over 30 000 participants aged 50–64 years underwent examinations of cardiopulmonary risk factors and provided information regarding lifestyle factors and socioeconomic conditions. Using the personal identity number, we linked SCAPIS data to nationwide administrative and health registers. From the National Patient Register, which comprises physician-assigned diagnoses based on the International Classification of Diseases, 10th Revision, Swedish Edition (ICD-10-SE) for all hospital admissions in Sweden, we obtained information about hospitalisation for COVID-19. From the Cause of Death register, we obtained data about vital status, and date and cause of death. From SmiNet,[10] a reporting system for infectious diseases administered by the Public Health Agency, we obtained information about laboratory-confirmed cases of COVID-19, including those not leading to hospitalisation or death.

### Outcomes, exposures, and study period
The main outcome was a composite of hospitalisation due to COVID-19 and death due to COVID-19. Hospitalisation for COVID-19 was defined as hospital admission with laboratory-confirmed COVID-19 as the primary diagnosis (ICD-10-SE code U07.1 (COVID-19, virus identified)).[11] Death due to COVID-19 was defined as death where U07.1 was specified as the underlying cause of death. Follow-up was from 31 January 2020 (the first laboratory-confirmed case of COVID-19 in Sweden) through 12 September 2020.

We included variables in the SCAPIS data set (described in detail in online supplemental table 1) that we hypothesised could be associated with hospitalisation or death due to COVID-19: these included sociodemographic variables (age, sex, place of birth and education) and cardiometabolic risk factors: diabetes status (normoglycaemia, pre-diabetes, diabetes); weight status (normal (body mass index (BMI)<25 kg/m$^2$), overweight (BMI ≥25 kg/m$^2$ to<30 kg/m$^2$), obesity (BMI ≥30 kg/m$^2$)); current smoking, waist-hip ratio, blood pressure status (normotension, grade 1 hypertension, grade 2 hypertension); systolic and diastolic blood pressure; coronary artery calcium score; total cholesterol; low density lipoprotein (LDL) cholesterol; high density lipoprotein (HDL) cholesterol; glycated haemoglobin and creatinine. Few (≤2%) participants had established cardiovascular disease, including coronary heart disease, stroke and heart failure; therefore, we did not assess these variables.

### Study population
We included all 30 154 participants in SCAPIS. We excluded 2 participants who had missing data on vital status and 197 participants who died before 31 January 2020. The study population included 29 955 participants.

### Statistical analysis
Analyses were performed in Stata V.16.0. We described study participants with respect to the selected variables, separately among those who did not experience hospitalisation or death due to COVID-19 and among those who did and assessed differences between the groups using the $\chi^2$ test for categorical variables and the t-test for continuous variables. For each investigated variable separately, we used logistic regression to assess its association with hospitalisation or death due to COVID-19, adjusting for sociodemographic variables including age (continuous variable), sex, place of birth (outside of Sweden vs Sweden) and education (high school or less, vocational, university).[12] Analyses were performed among those with complete data for all variables included in the model. The proportion of missing data for the analysed variables are shown in online supplemental table 1 and ranged between 0% and 5% (0%–8% in the main analyses using those with complete data on the variables assessed and adjusted for); the exception was time spent sedentary per day for which 34.9% of the participants had missing data. In post-hoc sensitivity analyses, we performed the main analyses with more than 5% missing data (those assessing waist-hip ratio, time spent sedentary per day and coronary artery calcium score) using multiple imputation (10 imputed data sets) created with chained equations.

The relationship between the investigated variables and hospitalisation or death due to COVID-19 may not only represent the risk of worse COVID-19 outcomes in individuals exposed to SARS-CoV-2 but also the risk of exposure to SARS-CoV-2. We therefore performed an additional analysis restricted to the subgroup of participants (n=299) with a laboratory-confirmed COVID-19 diagnosis as recorded in SmiNet, the National Patient Register or the Cause of Death Register (ie, participants with a laboratory-confirmed of COVID-19, including both those who did and did not experience hospitalisation or death due to COVID-19). In this subgroup, we used logistic regression to assess the association of each of the cardiometabolic risk factors (separately) with hospitalisation or death due to COVID-19. ORs whose 95% CI did not overlap 1 were considered as statistically significant.

### Patient involvement
No patients were involved in setting the research question, nor in the design, conduct or interpretation of the study. The study findings are planned to be disseminated through the SCAPIS website.

**Table 1** Characteristics of SCAPIS participants by experience of hospitalisation or death due to COVID-19 between 31 January and 12 September 2020

| | Hospitalisation or death due to COVID-19 | | |
| --- | --- | --- | --- |
| | No (n=29886) | Yes (n=69) | P value |
| Age, mean (SD) | 61.2 (4.5) | 61.9 (4.8) | 0.226 |
| Men | 14483 (48.5) | 52 (75.4) | <0.001 |
| Born outside of Sweden | 4725 (16.3) | 22 (32.4) | <0.001 |
| Education | | | |
| High school or less | 2700 (9.3) | 12 (17.9) | |
| Vocational | 13218 (45.5) | 31 (46.3) | |
| University | 13125 (45.2) | 24 (35.8) | 0.036 |
| Weight status | | | |
| Normal weight | 10738 (35.9) | 8 (11.6) | |
| Overweight | 12814 (42.9) | 36 (52.2) | |
| Obesity | 6332 (21.2) | 25 (36.2) | <0.001 |
| Body mass index in kg/m$^2$, mean (SD) | 27.0 (4.5) | 30.0 (5.3) | <0.001 |
| Diabetes status | | | |
| Normoglycaemia | 22774 (76.6) | 32 (46.4) | |
| Pre-diabetes | 4727 (15.9) | 20 (29.0) | |
| Diabetes | 2240 (7.5) | 17 (24.6) | <0.001 |
| Waist-hip ratio, mean (SD) | 0.92 (0.09) | 0.98 (0.09) | <0.001 |
| Current smoking | 3785 (13.1) | 6 (9) | 0.317 |
| Sedentary time per day in hours, mean (SD) | 6.9 (3.5) | 8.3 (4.5) | 0.009 |
| Blood pressure measurement | | | |
| Normotensive | 23285 (78.3) | 50 (72.5) | |
| Grade 1 hypertension | 5129 (17.3) | 12 (17.4) | |
| Grade 2 hypertension | 1319 (4.4) | 7 (10.1) | 0.069 |
| Systolic blood pressure, mean (SD) | 125.9 (17.0) | 133.5 (19.5) | <0.001 |
| Diastolic blood pressure, mean (SD) | 77.5 (10.5) | 81.0 (12.4) | 0.006 |
| Triglycerides in mmol/L, mean (SD) | 1.2 (0.8) | 1.6 (1.1) | <0.001 |
| HDL in mmol/L, mean (SD) | 1.6 (0.5) | 1.3 (0.4) | <0.001 |
| Total cholesterol in mmol/L, mean (SD) | 5.5 (1.1) | 5.3 (1.0) | 0.068 |
| LDL cholesterol in mmol/L, mean (SD) | 3.4 (1.0) | 3.3 (0.9) | 0.360 |
| Glycated haemoglobin in mmol/mol, mean (SD) | 36.6 (6.4) | 39.7 (9.3) | <0.001 |
| Creatinine in µmol/L, mean (SD) | 77.7 (16.4) | 86.2 (37.9) | <0.001 |
| Coronary artery calcium score, mean (SD) | 61.6 (229.5) | 90.2 (199.6) | 0.053* |

Numbers are shown in N (%) unless otherwise indicated.
*Because 17016 of the participants had a score of 0, the p value calculated using the $\chi^2$ with five categories: 0 and quartiles of score among participants with a score of >0.
HDL, high density lipoprotein; LDL, low density lipoprotein; SCAPIS, Swedish CArdioPulmonary bioImage Study.

## RESULTS

Characteristics of the 29955 study participants are shown in table 1. Fourteen thousand five hundred thirty-five (48.5%) were men. Mean (SD) age was 61.2 (4.5) years. Sixty-nine (0.2%) of the participants were hospitalised or died due to COVID-19. Those who experienced (n=69) versus did not experience (n=29886) hospitalisation or death due to COVID-19 were slightly older (61.9 years vs 61.2 years) and more likely to be men (75.4% vs 48.5%),

born outside of Sweden (32.4% vs 16.3%) and to have education of high school or less (17.9% vs 9.3%).

The results of the logistic regression assessing the association between each of the selected cardiometabolic risk factors and hospitalisation or death due to COVID-19 (adjusted for age, sex, place of birth and education) are shown in table 2. Significant associations were observed for overweight and obesity, higher body mass index, pre-diabetes, diabetes, higher waist-hip ratio, more time

**Table 2** OR (95% CI)[*], adjusted for age, sex, place of birth and education, for the association of selected variables with hospitalisation or death due to COVID-19 in the main and additional analyses

| | Main analysis (n=29 955) | Additional analysis restricted to those with laboratory-confirmed COVID-19 diagnosis (n=299) |
|---|---|---|
| Age per year | 1.03 (0.97 to 1.08) | 1.11 (1.04 to 1.20) |
| Men | 3.39 (1.93 to 5.95) | 6.62 (3.45 to 12.71) |
| Born outside of Sweden | 2.40 (1.43 to 4.03) | 1.59 (0.80 to 3.18) |
| Education† | | |
| High school or less | 1.00 (ref) | 1.00 (ref) |
| Vocational | 0.58 (0.30 to 1.14) | 0.41 (0.16 to 1.09) |
| University | 0.49 (0.24 to 0.99) | 0.26 (0.10 to 0.70) |
| Weight status | | |
| Normal weight | 1.00 (ref) | 1.00 (ref) |
| Overweight | 2.73 (1.25 to 5.94) | 3.41 (1.41 to 8.27) |
| Obesity | 4.09 (1.82 to 9.18) | 4.86 (1.86 to 12.71) |
| Body mass index per 5 kg/m$^2$ increase | 1.77 (1.43 to 2.19) | 2.05 (1.44 to 2.92) |
| Diabetes status | | |
| Normoglycaemia | 1.00 (ref) | 1.00 (ref) |
| Pre-diabetes | 2.56 (1.44 to 4.55) | 3.72 (1.68 to 8.28) |
| Diabetes | 3.96 (2.13 to 7.36) | 5.12 (1.95 to 13.42) |
| Waist-hip ratio per SD increase | 1.55 (1.20 to 2.00) | 1.92 (1.20 to 3.06) |
| Sedentary time per day per hour increase | 1.10 (1.02 to 1.17) | 1.13 (1.02 to 1.26) |
| Blood pressure measurement | | |
| Normotensive | 1.00 (ref) | 1.00 (ref) |
| Grade 1 hypertension | 1.03 (0.55 to 1.96) | 0.95 (0.41 to 2.17) |
| Grade 2 hypertension | 2.44 (1.10 to 5.44) | 4.18 (1.07 to 16.32) |
| Systolic blood pressure per 10 mm Hg increase | 1.22 (1.07 to 1.40) | 1.21 (1.01 to 1.44) |
| Diastolic blood pressure per 10 mm Hg increase | 1.31 (1.05 to 1.64) | 1.30 (0.97 to 1.73) |
| Current smoking | 0.56 (0.24 to 1.30) | 0.87 (0.28 to 2.71) |
| Triglycerides per mmol/L increase | 1.10 (1.00 to 1.21) | 1.34 (0.98 to 1.85) |
| HDL per mmol/L increase | 0.33 (0.17 to 0.65) | 0.26 (0.10 to 0.66) |
| Total cholesterol per mmol/L increase | 0.90 (0.71 to 1.13) | 0.80 (0.59 to 1.10) |
| LDL cholesterol per mmol/L increase | 0.90 (0.69 to 1.15) | 0.83 (0.59 to 1.17) |
| Glycated haemoglobin per mmol/mol increase | 1.03 (1.01 to 1.05) | 1.04 (1.00 to 1.09) |
| Creatinine per 10 µmol/L increase | 1.05 (1.00 to 1.10) | 1.04 (0.88 to 1.23) |
| Coronary artery calcium score per 10 units increase | 1.00 (0.99 to 1.01) | 1.00 (0.98 to 1.02) |

*Reference group for binary variables (yes/no) is 'no'.
†Adjusted for sex and place of birth.
HDL, high density lipoprotein; LDL, low density lipoprotein.

spent sedentary per day, grade 2 hypertension, as well as higher systolic blood pressure, diastolic blood pressure, triglycerides and glycated haemoglobin and lower HDL cholesterol. Significant associations were not observed for grade 1 hypertension, current smoking, total cholesterol, LDL cholesterol, creatinine and coronary artery calcium score. In the post-hoc sensitivity analyses using multiple imputation for logistic regression models with over 5% missing data, the results were similar to those in the main analyses: waist-hip ratio (OR per SD increase 1.54 (95% CI 1.20 to 1.98), time spent sedentary per day (OR per hour increase 1.09 (95% CI 1.01 to 1.18) and coronary

artery calcium score (OR per 10 units increase 1.00 (95% CI 0.99 to 1.01))

Population characteristics for the additional analysis including the 299 participants with laboratory confirmed COVID-19 are shown in online supplemental table 2. Compared with the total study population, those with a laboratory-confirmed diagnosis of COVID-19 were less likely to be men and to be current smokers and more likely to be born outside of Sweden. The results of the logistic regression analyses are shown in table 2. The findings were largely similar to those of the main analysis.

## DISCUSSION

We assessed the association of cardiometabolic risk factors with risk of hospitalisation or death due to COVID-19 during the first wave of the pandemic in a Swedish population-based cohort with nearly 30 000 participants aged 52–72 years in 2020.

In analyses adjusted for age, sex, place of birth and education, we found that several cardiometabolic risk factors were associated with an increased risk of hospitalisation or death due to COVID-19. Significant associations were found for metabolic risk factors, including overweight, obesity, pre-diabetes, diabetes and higher waist-hip ratio and glycated haemoglobin; findings that are in line with previous studies on these or related risk factors.[7 8 13] While significant associations were observed for grade 2 hypertension and for systolic and diastolic blood pressure when analysed as continuous variables, grade 1 hypertension was not associated with an increased risk. Moreover, while lower HDL cholesterol and higher triglycerides were associated with hospitalisation or death due to COVID-19, such associations were not observed for LDL cholesterol or total cholesterol. Associations with hospitalisation or death due to COVID-19 were also not observed for current smoking and a higher coronary artery calcium score. Previous studies on the association of smoking, hypertension, lipid levels and coronary artery calcium score with COVID-19 outcomes have yielded mixed findings.[8 14–18]

Strengths of our study include the use of a large sample from the general population who had undergone assessment of cardiometabolic risk factors in the years preceding the pandemic. Our study has limitations. Although the SCAPIS cohort includes detailed data on cardiometabolic risk factors from a large number of participants, the limited number of COVID-19 cases leading to hospitalisation or death resulted in wide CIs for some of the analyses. Moreover, during the first months of the pandemic, laboratory testing for COVID-19 was not widely performed and predominantly focused on healthcare professionals and hospitalised patients.[19] As such, although the findings of our analyses restricted to those with a laboratory-confirmed COVID-19 diagnosis were similar to those of our main analyses (although the analyses were based on a small sample of 299 participants), we could not assess to what extent the observed associations may reflect the

relationship with exposure to SARS-CoV-2 as compared with the risk of hospitalisation or death due to COVID-19 among those who have been exposed to the virus. Updated analyses may be performed as data on broader testing for COVID-19 and more cases of hospitalisation or death due to COVID-19 become available. Finally, the study included individuals in a limited age range.

## CONCLUSION

In this study, large population-based cohort from the general population, several cardiometabolic risk factors were associated with hospitalisation or death due to COVID-19.

**Author affiliations**
[1]Functional Area of Emergency Medicine, Karolinska University Hospital Huddinge, Stockholm, Sweden
[2]Department of Clinical Sciences, Danderyd University Hospital, Karolinska Institutet, Stockholm, Sweden
[3]Clinical Physiology, Sahlgrenska University Hospital, Gothenburg, Sweden
[4]Department of Molecular and Clinical Medicine, Sahlgrenska Academy, University of Gothenburg, Gothenburg, Sweden
[5]Department of Public Health and Clinical Medicine, Section of Medicine, Umeå University, Umeå, Sweden
[6]Department of Clinical Sciences in Malmö, Lund University, Malmö, Sweden
[7]CMIV, Centre of Medical Image Science and Visualization, Linköping University, Linköping, Sweden
[8]Department of Health, Medicine and Caring Sciences, Linköping University, Linköping, Sweden
[9]Department of Medical Sciences, Molecular Epidemiology and Science for Life Laboratory, Uppsala University, Uppsala, Sweden
[10]Department of Infectious Diseases, Institute of Biomedicine, Sahlgrenska Academy, University of Gothenburg, Gothenburg, Sweden
[11]Department of Infectious Diseases, Region Västra Götaland, Sahlgrenska University Hospital, Gothenburg, Sweden
[12]Department of Medical Sciences, Respiratory-, Allergy- and Sleep Research, Uppsala University, Uppsala, Sweden
[13]Department of Medical Sciences, Clinical Epidemiology, Uppsala University, Uppsala, Sweden
[14]Respiratory Medicine Unit, Department of Medicine Solna and Center for Molecular Medicine, Karolinska Institutet, Stockholm, Stockholm, Sweden
[15]Department of Respiratory Medicine and Allergy, Karolinska University Hospital Solna, Stockholm, Sweden
[16]The George Institute for Global Health, University of New South Wales, Sydney, New South Wales, Australia
[17]Department of Medicine Huddinge H7, Karolinska Institutet, Karolinska University Hospital, Stockholm, Sweden
[18]Clinical Epidemiology Division, Department of Medicine Solna, Karolinska Institutet, Stockholm, Sweden

**Contributors** PU had full access to all the data in the study and took responsibility for the integrity of the data and the accuracy of the data analysis. PT, TJ, TF, CJ, MS, DPA and PU developed the research design. PT, TJ, GB, AB, GE, JE, TF, MG, CJ, LL, MS, JS, SS, SZ, CJÖ, DPA and PU contributed to the acquisition, analysis or interpretation of data. DPA and PU drafted first draft of the manuscript. PT, TJ, GB, AB, GE, JE, TF, MG, CJ, LL, MS, JS, SS, SZ, CJÖ, DPA and PU critically reviewed and revised the manuscript for important intellectual content and approved the manuscript. PU performed the statistical analysis. TJ and PU obtained funding.

**Funding** The study was supported by a grant from the Swedish Heart-Lung Foundation (grant number 20200491). PU was supported by grants from the Swedish Heart-Lung Foundation and the Swedish Society for Medical Research. DPPA was supported by grants from CIMED (grant number 20180855). The main funding body of The Swedish CArdioPulmonary bioImage Study (SCAPIS) is the Swedish Heart-Lung Foundation. The study is also funded by the Knut and Alice

Wallenberg Foundation, the Swedish Research Council and VINNOVA (Sweden's Innovation agency), the University of Gothenburg and Sahlgrenska University Hospital, Karolinska Institute and Stockholm county council, Linköping University and University Hospital, Lund University and Skåne University Hospital, Umeå University and University Hospital, Uppsala University and University Hospital.

**Disclaimer** The funding sources had no role in the design and conduct of the study; collection, management, analysis and interpretation of the data; preparation, review or approval of the manuscript and decision to submit the manuscript for publication.

**Competing interests** None declared.

**Patient consent for publication** Not required.

**Ethics approval** The study was approved by the Ethical Review Board in Umeå, Sweden (reference numbers: 2010-228-31M and 2020-02668).

**Provenance and peer review** Not commissioned; externally peer reviewed.

**Data availability statement** No data are available.

**ORCID iDs**
Tove Fall http://orcid.org/0000-0003-2071-5866
Christer Janson http://orcid.org/0000-0001-5093-6980
Stefan Söderberg http://orcid.org/0000-0001-9225-1306
Daniel Peter Andersson http://orcid.org/0000-0003-4655-4837
Peter Ueda http://orcid.org/0000-0002-3275-8743

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
