## [Reviewer comments · BMJ Open]

ARTICLE DETAILS

TITLE (PROVISIONAL)	Association of cardiometabolic risk factors with hospitalization or death due to COVID-19: population-based cohort study in Sweden (SCAPIS)
AUTHORS	Tornhammar, Per; Jernberg, Tomas; Bergström, Göran; Blomberg, Anders; Engström, Gunnar; Engvall, Jan; Fall, Tove; Gisslén, Magnus; Janson, Christer; Lind, Lars; Skold, Magnus; Sundström, Johan; Söderberg, Stefan; Zaigham, Suneela; Östgren, Carl Johan; Andersson, Daniel Peter; Ueda, Peter;

VERSION 1 – REVIEW

REVIEWER	Timoteo, Ana Teresa Ctr Hosp Lisboa Cent
REVIEW RETURNED	28-Mar-2021

GENERAL COMMENTS	The authors of the present manuscript sought to assess a very relevant subject: the association of cardiometabolic risk factors with hospitalization or death due to Covid-19 in the general population. For the study, they used data from a Swedish population-based cohort from the Swedish Cardiopulmonary bioimage study (SCAPIS) and included 29,955 participants. Exposures Cardiometabolic risk factors were assessed between 2014 and 2018 and the main outcome measures were hospitalization or death due to Covid-19, as registered in nationwide registers from January 31 through September 12, 2020. The associations of cardiometabolic risk factors with the outcome were assessed using logistic regression and were adjusted for age, sex, birthplace, and education. In this cohort, 69 participants (0.2%) experienced hospitalization or death due to Covid-19. Statistically significant associations between baseline factors and outcome included overweight, prediabetes, diabetes, sedentary time, grade 2 hypertension and high-density lipoprotein cholesterol. No association was observed for grade 1 hypertension, current smoking, total cholesterol, low density lipoprotein cholesterol and coronary artery calcium score. From this data, they concluded that in a large population-based sample from the general population, several cardiometabolic risk factors were associated with hospitalization or death due to Covid19. This is an interesting and relevant study, on a very up to date subject. Data from China and other countries had already suggested that patients with cardiovascular disease were at increased risk of adverse outcomes in the context of COV119 infection. This study showed that cardiometabolic risk factors were also associated with increased risk. This is particularly important
---

	because there is an urgent need to identify and prioritize patients for COVID19 vaccination. It is of particularly importance the findings that cardiometabolic risk factors are more relevant in terms of worse outcome in COVID19 patients when compared to conventional cardiovascular risk factors, such as smoking and LDL-cholesterol. In general, methodology used was appropriate for the objective of the present study. They also performed an additional analysis in 299 patients with laboratory confirmed COVID19 to confirm the results obtained in the general cohort. References are also recent. Table 1 should have a p-value column to better appreciate significant differences between groups. The results of logistic regression analysis are extensively repeated in the text. Data is already presented in the Table 2 and only a summary should be in the main text. Additional limitation:  - Only patients from 50 to 64 years of age were included in the study and for that reason, a substantial proportion of the population was not assessed, and the study did not provide information. - COVID19 testing was only available in a very limited number of patients and this might be an important cause for bias. However, the tested group (299 patients) showed similar results. However, the authors should present a table with a head-to-head comparison of the overall population and the tested patients to see if there are any significant different. Is this sub-group representative? - Also, the 299 cohort is a very small sample, and this has important consequences regarding statistical power that should be addressed by the authors.
--	---

REVIEWER	Lebovic, Gerald St. Michael's Hospital, Applied Health Research Centre, Li Ka Shing Knowledge Institute
REVIEW RETURNED	26-Apr-2021

GENERAL COMMENTS	Thank you for giving me the opportunity to review this paper. Comments:  1) It seems like SCAPIS contains cardiovascular and some other data on comorbidities. What limitations exist due to the data that is not collected in here (e.g. other comorbidities etc.) 2) Is the main outcome a composite of hospitalization and/or death or each one separate? 3) For those missing data on vital status did you perform a sensitivity analysis? 4) Statistical analysis line 41-43: "Analyses were performed among those with complete data for each analyzed variable" – what does this mean. Was any imputation considered? How much data was dropped because of this? 5) There are a large number of variables included in the model yet there are only 69 hospitalizations or death. This strikes me as overfitting the model. The rule of thumb of 10 events per variable may not necessarily always be adhered to but based on the size
---

	of the CIS I would say there may be some overfitting. Was any internal validation done? However, even with my concern above, my main concern is that the cohort selected is not appropriate to address the research questions. The primary analysis uses all subjects (with complete data). Having a Covid-19 positive test is not required. Therefore this analysis only addresses having a hospitalization or death DURING the Covid-19 pandemic it CANNOT say anything about the odds of associations with risk of death or hospitalizations “due to Covid”/ The secondary analysis restricts the cohort to those that tested positive for Covid-19 and in my opinion that is the valid analysis and proper cohort. Minor: Abstract: Line32 – Jan 31 should have 2020 after it Page 8 should start with Introduction header Page 8 line 8 remove word an
--	--

VERSION 1 – AUTHOR RESPONSE

Reviewer: 1

Dr. Ana Teresa Timoteo, Ctr Hosp Lisboa Cent

1.The authors of the present manuscript sought to assess a very relevant subject: the association of cardiometabolic risk factors with hospitalization or death due to Covid-19 in the general population.

Thank you for the valuable comments which have led to further improvements of our manuscript.

2.For the study, they used data from a Swedish population-based cohort from the Swedish Cardiopulmonary bioimage study (SCAPIS) and included 29,955 participants. Exposures Cardiometabolic risk factors were assessed between 2014 and 2018 and the main outcome measures were hospitalization or death due to Covid-19, as registered in nationwide registers from January 31 through September 12, 2020. The associations of cardiometabolic risk factors with the outcome were assessed using logistic regression and were adjusted for age, sex, birthplace, and education. In this cohort, 69 participants (0.2%) experienced hospitalization or death due to Covid-19. Statistically significant associations between baseline factors and outcome included overweight, prediabetes, diabetes, sedentary time, grade 2 hypertension and high-density lipoprotein cholesterol. No association was observed for grade 1 hypertension, current smoking, total cholesterol, low density lipoprotein cholesterol and coronary artery calcium score. From this data, they concluded that in a large population-based sample from the general population, several cardiometabolic risk factors were associated with hospitalization or death due to Covid19.

This is an interesting and relevant study, on a very up to date subject. Data from China and other countries had already suggested that patients with cardiovascular disease were at increased risk of adverse outcomes in the context of COVI19 infection. This study showed that cardiometabolic risk factors were also associated with increased risk. This is particularly important because there is an urgent need to identify and prioritize patients for COVID19 vaccination. It is of particularly importance the findings that cardiometabolic risk factors are more relevant in terms of worse outcome in COVID19 patients when compared to conventional cardiovascular risk factors, such as smoking and LDL-cholesterol.

Thank you.

3. In general, methodology used was appropriate for the objective of the present study. They also performed an additional analysis in 299 patients with laboratory confirmed COVID-19 to confirm the results obtained in the general cohort. References are also recent.

Thank you.

4. Table 1 should have a p-value column to better appreciate significant differences between groups.

To Table 1 we have now added p-values for group differences between those experiencing hospitalization or death due to Covid-19 vs those who did not using the Chi-squared test for categorical variables and the t-test for continuous variables. We have described this in the methods.

Methods:

“We described study participants with respect to the selected variables, separately among those who did not experience hospitalization or death due to Covid-19 and among those who did and assessed differences between the groups using the Chi-squared test for categorical variables and the t-test for continuous variables.”

Table 1 Characteristics of SCAPIS participants by experience of hospitalization or death due to Covid-19 between January 31 and September 12, 2020. Numbers are shown in n (%) unless otherwise indicated.

	Hospitalization or death due to Covid-19		P-value
	No (n=29,866)	Yes (n=69)	
Age, mean (SD)	61.2 (4.5)	61.9 (4.8)	0.226
Men	14483 (48.5)	52 (75.4)	<0.001
Born outside of Sweden	4725 (16.3)	22 (32.4)	<0.001
Education			
High School or less	2700 (9.3)	12 (17.9)	
Vocational	13218 (45.5)	31 (46.3)	
University	13125 (45.2)	24 (35.8)	0.036

Weight status			
Normal weight	10738 (35.9)	8 (11.6)	
Overweight	12814 (42.9)	36 (52.2)	
Obesity	6332 (21.2)	25 (36.2)	<0.001
Body mass index in kg/m ² , mean (SD)	27.0 (4.5)	30.0 (5.3)	<0.001
Diabetes status			
Normoglycemia	22774 (76.6)	32 (46.4)	
Prediabetes	4727 (15.9)	20 (29.0)	
Diabetes	2240 (7.5)	17 (24.6)	<0.001
Waist-hip ratio, mean (SD)	0.92 (0.09)	0.98 (0.09)	<0.001
Current smoking	3785 (13.1)	6 (9)	0.317
Sedentary time per day in hours, mean (SD)	6.9 (3.5)	8.3 (4.5)	0.009
Blood pressure measurement			
Normotensive	23285 (78.3)	50 (72.5)	
Grade 1 hypertension	5129 (17.3)	12 (17.4)	
Grade 2 hypertension	1319 (4.4)	7 (10.1)	0.069
Systolic blood pressure, mean (SD)	125.9 (17.0)	133.5 (19.5)	<0.001
Diastolic blood pressure, mean (SD)	77.5 (10.5)	81.0 (12.4)	0.006

Triglycerides in mmol/L, mean (SD)	1.2 (0.8)	1.6 (1.1)	<0.001
HDL in mmol/L, mean (SD)	1.6 (0.5)	1.3 (0.4)	<0.001
Total cholesterol in mmol/L, mean (SD)	5.5 (1.1)	5.3 (1.0)	0.068
LDL cholesterol in mmol/L, mean (SD)	3.4 (1.0)	3.3 (0.9)	0.360
Glycated hemoglobin in mmol/mol, mean (SD)	36.6 (6.4)	39.7 (9.3)	<0.001
Creatinine in μ mol/L, mean (SD)	77.7 (16.4)	86.2 (37.9)	<0.001
Coronary artery calcium score, mean (SD)	61.6 (229.5)	90.2 (199.6)	0.316

5. The results of logistic regression analysis are extensively repeated in the text. Data is already presented in the Table 2 and only a summary should be in the main text.

We have removed the odds ratios from the text in the Results section and only provide a summary of the findings.

“The results of the logistic regression assessing the association between each of the selected cardiometabolic risk factors and hospitalization or death due to Covid-19 (adjusted for age, sex, place of birth and education) are shown in Table 2. Significant associations were observed for overweight and obesity, higher body mass index, prediabetes, diabetes, higher waist-hip ratio, more time spent sedentary per day, grade 2 hypertension, as well as higher systolic blood pressure, diastolic blood pressure, triglycerides and glycated hemoglobin, and lower HDL cholesterol. Significant associations were not observed for grade 1 hypertension, current smoking, total cholesterol, LDL cholesterol, creatinine and coronary artery calcium score.”

6. Additional limitation:

- Only patients from 50 to 64 years of age were included in the study and for that reason, a substantial proportion of the population was not assessed, and the study did not provide information.

We have added this limitation to the limitations section of the Discussion:

“Finally, the study included individuals in a limited age range.”

- COVID19 testing was only available in a very limited number of patients and this might be an important cause for bias. However, the tested group (299 patients) showed similar results. However, the authors

should present a table with a head-to-head comparison of the overall population and the tested patients to see if there are any significant different. Is this sub-group representative?

Thank you for this suggestion. In Supplementary Table 2, we now present more comprehensive participant characteristics including those who had no laboratory-confirmed diagnosis of Covid-19; those who had a laboratory-confirmed diagnosis but did not experience hospitalization or death and those who were hospitalized or died due to Covid-19. We would like to highlight the comparison of those without a laboratory-confirmed diagnosis with those who had a laboratory-confirmed diagnosis but who were not hospitalized or died (we considered this comparison as more important because those who experienced hospitalization/death were likely to have been tested due to their severe Covid-19 outcome and would thus have characteristics that made them susceptible to adverse outcomes in Covid-19). Those with a laboratory-confirmed Covid-19 diagnosis but who were not hospitalized or died differed vs those without a laboratory-confirmed diagnosis with respect to several variables. For example, they were slightly younger, less likely to be men and more likely to be born outside of Sweden; they also tended to have lower levels of some of the cardiometabolic risk factors.

We have now mentioned this in the Results.

“Compared to those without a laboratory-confirmed diagnosis of Covid-19, those with a laboratory-confirmed diagnosis but who did not experience hospitalization or death due to Covid-19 were less likely to be men, be current smokers and to have diabetes and more likely to be born outside of Sweden and to have a university education and a BMI within the normal range.”

Supplementary Table 2 Characteristics of SCAPIS participants by their status of laboratory-confirmed diagnosis Covid-19 diagnosis between January 31 and September 12, 2020. Numbers are shown in n (%) unless otherwise indicated.

	No laboratory-confirmed diagnosis of Covid-19	Laboratory-confirmed diagnosis of Covid-19	
		Hospitalization or death due to Covid-19	
n		No	Yes
	29656	230	69
Age, mean (SD)	61.2 (4.5)	59.9 (4.0)	61.9 (4.8)
Men	14408 (48.6)	75 (32.6)	52 (75.4)

Born outside of Sweden	4676 (16.2)	49 (21.7)	22 (32.4)
Education			
High School or less	2686 (9.3)	14 (6.2)	12 (17.9)
Vocational	13120 (45.5)	98 (43.4)	31 (46.3)
University	13011 (45.2)	114 (50.4)	24 (35.8)
Weight status			
Normal weight	10643 (35.9)	95 (41.3)	8 (11.6)
Overweight	12723 (42.9)	91 (39.6)	36 (52.2)
Obesity	6288 (21.2)	44 (19.1)	25 (36.2)
Body mass index in kg/m ² , mean (SD)	27.0 (4.5)	26.5 (4.2)	30 (5.3)
Diabetes status			
Normoglycemia	22582 (76.5)	192 (83.5)	32 (46.4)
Prediabetes	4701 (15.9)	26 (11.3)	20 (29)
Diabetes	2228 (7.5)	12 (5.2)	17 (24.6)
Waist-hip ratio, mean (SD)	0.90 (0.1)	0.90 (0.08)	0.98 (0.09)
Current smoking	3768 (13.1)	17 (7.7)	6 (9.0)
Sedentary time per day in hours, mean (SD)	6.9 (3.5)	6.1 (3.4)	8.3 (4.5)

Blood pressure level			
Normotensive	23100 (78.3)	185 (81.1)	50 (72.5)
Grade 1 hypertension	5093 (17.3)	36 (15.8)	12 (17.4)
Grade 2 hypertension	1312 (4.4)	7 (3.1)	7 (10.1)
Systolic blood pressure, mean (SD)	125.9 (17.0)	124.2 (16.9)	133.5 (19.5)
Diastolic blood pressure, mean (SD)	77.5 (10.5)	76.6 (10.0)	81 (12.4)
Triglycerides in mmol/L, mean (SD)	1.2 (0.8)	1.2 (0.8)	1.6 (1.1)
HDL in mmol/L, mean (SD)	1.6 (0.5)	1.7 (0.5)	1.3 (0.4)
Total cholesterol in mmol/L, mean (SD)	5.5 (1.1)	5.6 (1.0)	5.3 (1.0)
LDL cholesterol in mmol/L, mean (SD)	3.4 (1.0)	3.6 (0.9)	3.3 (0.9)
Glycated hemoglobin in mmol/mol, mean (SD)	36.6 (6.5)	35.9 (5.6)	39.7 (9.3)
Creatinine in μ mol/L, mean (SD)	77.7 (16.4)	74.6 (14.4)	86.2 (37.9)
Coronary artery calcium score, mean (SD)	61.8 (230.1)	24.4 (119.9)	90.2 (199.6)

- Also, the 299 cohort is a very small sample, and this has important consequences regarding statistical power that should be addressed by the authors.

We have now clarified this in the limitations section of the Discussion:

“As such, although the findings of our analyses restricted to those with a laboratory-confirmed Covid-19 diagnosis were similar to those of our main analyses (although the analyses were based on a small sample of 299 participants), we could not assess to what extent the observed associations may reflect the relationship with exposure to SARS-CoV 2 as compared with the risk of hospitalization or death due to Covid-19 among those who have been exposed to the virus.”

Reviewer: 2

Dr. Gerald Lebovic, St. Michael's Hospital, Applied Health Research Centre

Comments to the Author:

Thank you for giving me the opportunity to review this paper.

Thank you for the valuable comments which have led to further improvements of our manuscript.

Comments:

1) It seems like SCAPIS contains cardiovascular and some other data on comorbidities. What limitations exist due to the data that is not collected in here (e.g. other comorbidities etc.)

Thank you for highlighting this. We agree other variables, including comorbidities such as established cardiovascular disease – e.g. history of myocardial infarction, stroke and heart failure - might be associated with risk of worse outcomes with Covid-19. Data on comorbidities were available in the SCAPIS dataset. However, SCAPIS participants were between 50 and 64 years when baseline data were collected and the proportion with established comorbidities were relatively low; specifically, established cardiovascular disease was rare. For example, in our study population only 2% had a history of coronary artery disease, 1% had a history of stroke and 0.5% had a history of heart failure. Therefore, we did not include these variables in our analyses. We have now explained this in the Methods section:

“Few ($\leq 2\%$) participants had established cardiovascular disease, including coronary heart disease, stroke and heart failure; therefore, we did not assess these variables.”

We also agree that the relationship between the investigated cardiometabolic variables and hospitalization or death due to Covid-19 may be partly attributable to other factors associated with both the cardiometabolic risk factor and the outcome. While we adjusted the analyses for age, sex, place of birth and education, our analyses did not aim to establish causality but to provide an overview of the associations with hospitalization or death due to Covid-19 to help identifying individuals at risk of worse outcomes in Covid-19.

2) Is the main outcome a composite of hospitalization and/or death or each one separate?

It is the composite of hospitalization or death due to Covid-19. We have now further clarified this in the Methods section:

“The main outcome was a composite of hospitalization due to Covid-19 and death due to Covid-19.”

Throughout the paper, we refer to the outcome as “hospitalization or death due to Covid-19” and we consider the outcome definition to be clearly described in the paper.

3) For those missing data on vital status did you perform a sensitivity analysis?

Of the 199 SCAPIS participants who were excluded, 197 were excluded because they had died before the start of the Covid-19 pandemic and only 2 individuals were excluded due to missing vital status. As such, we did not perform any sensitivity analyses for these individuals. We have now clarified this.

“We included all 30,154 participants in SCAPIS. We excluded 2 participants who had missing data on vital status and 197 participants who died before January 31, 2020.”

4) Statistical analysis line 41-43: “Analyses were performed among those with complete data for each analyzed variable” – what does this mean. Was any imputation considered? How much data was dropped because of this?

Study participants with no missing data for all variables included in a specific model was used. For example, in the logistic regression model assessing the association of systolic blood pressure with hospitalization or death due to Covid-19, only individuals with no missing data on systolic blood pressure as well as the variables adjusted for (age, sex, place of birth and education) were included. We have now clarified this further and added the n (%) with missing data for each variable in Supplementary Table 1 and the briefly described missing data in the Methods section. Given the small proportion of missingness (for all variables except sedentary time spent per day), we did not consider imputation.

Methods

“Analyses were performed among those with complete data for all variables included in the model. The proportion of missing data for the analyzed variables are shown in Supplementary Table 1 and ranged between 0 and 5%; the exception was time spent sedentary per day for which 34.9% of the participants had missing data.”

Supplementary Table 1

Supplementary Table 1 Definitions and categorization of selected variables.

Variable	Categorization	n (%) missing
Sociodemographic information		
Age	Continuous in years	0 (0)
Sex	1. Women 2. Men	0 (0)
Place of birth	1. Not born in Sweden 2. Born in Sweden	811 (2.7)

Education	 1. High school or less 2. Vocational education 3. University 	845 (2.8)
Cardiometabolic risk factors		
Diabetes status	 1. Normoglycemia 2. Prediabetes (fasting glucose [6.1-6.9 mmol/L or glycated hemoglobin \geq42 mmol/mol and $<$48 mmol/mol]) 3. Diabetes diagnosis by physician (self-reported in questionnaire) or glycated hemoglobin \geq48 mmol/mol. 	145 (0.5)
Glycated hemoglobin	Continuous in mmol/mol	152 (0.5)
Body mass index	Continuous in kg/m ²	2 (0)
Weight status	 1. Normal weight 2. Overweight 3. Obesity 	2 (0)
Waist-hip ratio	Continuous	1567 (5.2)
Systolic blood pressure	Continuous in mmHg	151 (0.5)
Diastolic blood pressure	Continuous in mmHg	153 (0.5)
Blood pressure level	Level as measured at inclusion in SCAPIS.  1. Normotensive (systolic blood pressure $<$140 mmHg and diastolic blood pressure $<$90 mmHg) 2. Grade 1 hypertension (Systolic blood pressure \geq140 mmHg and $<$160 mmHg or diastolic blood pressure \geq90 mmHg and $<$100 mmHg) 3. Grade 2 hypertension (systolic blood pressure \geq160 mmHg or diastolic blood pressure \geq100 mmHg). 	153 (0.5)
Current smoking	Self-reported in questionnaire.  1. No 2. Yes 	954 (3.2)

Time spent sedentary per day	Self-reported in questionnaire. Continuous (hours per day)	10441 (34.9)
Coronary artery calcium score	Continuous	1197 (4)
Total cholesterol	Continuous in mmol/L	82 (0.3)
Low-density lipoprotein (LDL) cholesterol	Continuous in mmol/L	218 (0.7)
HDL cholesterol	Continuous in mmol/L	85 (0.3)
Creatinine	Continuous in mikromol/L	61 (0.2)

a. n (%) missing values out of the total study population (n=29,955)

5) There are a large number of variables included in the model yet there are only 69 hospitalizations or death. This strikes me as overfitting the model. The rule of thumb of 10 events per variable may not necessarily always be adhered to but based on the size of the CIS I would say there may be some overfitting. Was any internal validation done?

We included only a limited number of important variables in the analyses: the logistic regression models assessed each cardiometabolic risk factor separately and were adjusted for age, sex, place of birth and education. Accordingly, there were around 10 events per variable. We did not include all the cardiometabolic risk factors in the same model.

We have now further clarified that the cardiometabolic risk factors were assessed separately.

Methods

“For each cardiometabolic risk factor separately, we used logistic regression to assess its association with hospitalization or death due to Covid-19, adjusting for sociodemographic variables including age (continuous variable), sex, place of birth (outside of Sweden vs Sweden) and education (high school or less, vocational, university).”

“In this subgroup, we used logistic regression to assess the association of each of the cardiometabolic risk factors (separately) with hospitalization or death due to Covid-19.”

Results

“The results of the logistic regression assessing the association between each of the selected cardiometabolic risk factors and hospitalization or death due to Covid-19 (adjusted for age, sex, place of birth and education) are shown in Table 2”

However, even with my concern above, my main concern is that the cohort selected is not appropriate to address the research questions. The primary analysis uses all subjects (with complete data). Having a Covid-19 positive test is not required. Therefore this analysis only addresses having a hospitalization or death DURING the Covid-19 pandemic it CANNOT say anything about the odds of associations with risk of death or hospitalizations “due to Covid”/ The secondary analysis restricts the cohort to those that tested positive for Covid-19 and in my opinion that is the valid analysis and proper cohort.

We agree that a limitation of our study was that we did not have complete data regarding which individuals had been exposed to SARS-CoV-2; thus we could not with certainty assess to what extent the observed associations may reflect the relationship with exposure to SARS-CoV-2 as compared with the risk of hospitalization or death due to Covid-19 among those who have been exposed to the virus.

Incomplete information about exposure to SARS-CoV-2 is a limitation shared by many previous population-based studies on risk factors for severe Covid-19. Importantly, these studies have nonetheless been crucial for identification of risk groups during the pandemic. For example, the study by Williamson et al using data from OpenSAFELY in England (Nature 2020)¹ had a similar design as our present study (although with a substantially larger sample size) and included *no* data regarding exposure to SARS-CoV-2. Other examples of studies with this limitation include McGurnaghan et al (Lancet Diabetes & Endocrinology 2020)², Clift et al (BMJ 2020)³ and Barron et al (Lancet Diabetes & Endocrinology 2020)⁴.

In the *Discussion* of our paper, we have clearly described the limitation of incomplete information about exposure to SARS-CoV-2. To address this limitation, we have also performed an additional analysis restricted to those with a laboratory-confirmed Covid-19 diagnosis (including cases that did not lead to hospitalization), although many cases were likely to be undiagnosed as testing was limited early in the pandemic. Nonetheless, the findings of this additional analysis were largely similar to those of our main analysis, indicating the bias due to differential exposure to SARS-CoV-2 across levels of cardiometabolic risk factors was limited.

Taken together, while the incomplete data on exposure to SARS-Cov2 is a limitation, population-based analyses of risk factors for severe Covid-19 are informative and important. Our study also has significant strengths, including the use of a large sample of almost 30 000 individuals from the general population who had undergone detailed assessment of cardiometabolic risk factors in the years preceding the pandemic and nationwide registers with complete capture of Covid-19 hospitalizations and death.

Minor:

Abstract:

Line32 – Jan 31 should have 2020 after it

We have now revised accordingly.

Page 8 should start with Introduction header

We have now revised accordingly.

Page 8 line 8 remove word an

We have now revised accordingly.

Reviewer: 1

Competing interests of Reviewer: None

Reviewer: 2

Competing interests of Reviewer: None declared

Additional changes made to the manuscript:

The variable current smoking was missing in Table 1 and Supplementary Table 2 in the previous version of the manuscript: this variable has now been added.

Due to an administrative error, three co-authors who fulfill the authorship criteria had not been included in the previous version of the manuscript. These authors have now been included.

References

1. Williamson EJ, Walker AJ, Bhaskaran K, et al. Factors associated with COVID-19-related death using OpenSAFELY. *Nature*. 2020;584(7821):430-436.
2. McGurnaghan SJ, Weir A, Bishop J, et al. Risks of and risk factors for COVID-19 disease in people with diabetes: a cohort study of the total population of Scotland. *Lancet Diabetes Endocrinol*. December 2020.
3. Clift AK, Coupland CAC, Keogh RH, et al. Living risk prediction algorithm (QCOVID) for risk of hospital admission and mortality from coronavirus 19 in adults: national derivation and validation cohort study. *BMJ*. 2020;371:m3731.
4. Barron E, Bakhai C, Kar P, et al. Associations of type 1 and type 2 diabetes with COVID-19-related mortality in England: a whole-population study. *Lancet Diabetes Endocrinol*. 2020;8(10):813-822.

VERSION 2 – REVIEW

REVIEWER	Timoteo, Ana Teresa Ctr Hosp Lisboa Cent
REVIEW RETURNED	03-Jul-2021

GENERAL COMMENTS	In general, my comments and the comments from the second reviewer were properly addressed by the authors. However, as I mentioned in my comments and also the second reviewer points out, the real population where direct conclusions can be drawn is the group that was tested for COVID19 (n=299). Supplemental table 2 does not show the right comparison between the total population and the 299 who had a COVID 19 testing. This should be the comparison to be carried out to see if the smaller group is really representative of the general population. This table should be presented in this format: overall study population and overall COVID19 tested population.
---

REVIEWER	Lebovic, Gerald St. Michael's Hospital, Applied Health Research Centre, Li Ka Shing Knowledge Institute
REVIEW RETURNED	22-Jul-2021

GENERAL COMMENTS	Thank you very much for your responses and for further clarification. I have one main concern. With respect to missing data you mention that no imputation was done as almost all variables used had very little data missing. I agree that this is true however if you include education, place of birth and another cardiometabolic variable, say coronary artery calcium, then if there is little overlap of the missing data you can potentially have approx. 9% of the data missing. I think you need to see in each model how many observations were dropped due to missing data and if > 5% should consider at least single imputation as a sensitivity analysis (ideally multiple imputation). I would simply state a sensitivity analysis was done and we found X.
---

VERSION 2 – AUTHOR RESPONSE

Reviewer: 1

Dr. Ana Teresa Timoteo, Ctr Hosp Lisboa Cent

Thank you for the comments on our manuscript. As suggested, we have now revised Supplemental Table 2 such that it shows the overall study population and the population with a laboratory-confirmed diagnosis of Covid-19. We have also revised the text in Results highlighting some of the differences in characteristics between the groups.

Supplementary Material

Supplementary Table 2 Characteristics of SCAPIS participants by their status of laboratory-confirmed diagnosis of Covid-19 between January 31 and September 12, 2020. Numbers are shown in n (%) unless otherwise indicated.

	Total study population	Laboratory-confirmed diagnosis of Covid-19
n		
	29955	299

Age, mean (SD)	61.2 (4.5)	60.3 (4.2)
Men	14535 (48.5)	127 (42.5)
Born outside of Sweden	4747 (16.3)	71 (24.1)
Education		
High School or less	2712 (9.3)	26 (8.9)
Vocational	13249 (45.5)	129 (44.0)
University	13149 (45.2)	138 (47.1)
Weight status		
Normal weight	10746 (35.9)	103 (34.4)
Overweight	12850 (42.9)	127 (42.5)
Obesity	6357 (21.2)	69 (23.1)
Body mass index in kg/m ² , mean (SD)	27.0 (4.5)	27.3 (4.7)
Diabetes status		
Normoglycemia	22806 (76.5)	224 (74.9)
Prediabetes	4747 (15.9)	46 (15.4)
Diabetes	2257 (7.6)	29 (9.7)
Waist-hip ratio, mean (SD)	0.9 (0.1)	0.9 (0.1)
Current smoking	3791 (13.1)	23 (8.0)

Sedentary time per day in hours, mean (SD)	6.9 (3.6)	6.6 (3.8)
Blood pressure level		
Normotensive	23335 (78.3)	235 (79.1)
Grade 1 hypertension	5141 (17.3)	48 (16.2)
Grade 2 hypertension	1326 (4.4)	14 (4.7)
Systolic blood pressure, mean (SD)	125.9 (17.0)	126.4 (17.9)
Diastolic blood pressure, mean (SD)	77.5 (10.5)	77.6 (10.7)
Triglycerides in mmol/L, mean (SD)	1.2 (0.8)	1.3 (0.9)
HDL in mmol/L, mean (SD)	1.6 (0.5)	1.6 (0.5)
Total cholesterol in mmol/L, mean (SD)	5.5 (1.1)	5.6 (1)
LDL cholesterol in mmol/L, mean (SD)	3.4 (1.0)	3.5 (0.9)
Glycated hemoglobin in mmol/mol, mean (SD)	36.6 (6.5)	36.8 (6.8)
Creatinine in μ mol/L, mean (SD)	77.7 (16.5)	77.3 (22.6)
Coronary artery calcium score, mean (SD)	61.6 (229.5)	39.5 (144.4)

Results

“Compared to the total study population, those with a laboratory-confirmed diagnosis of Covid-19 were less likely to be men and to be current smokers and more likely to be born outside of Sweden.”

Reviewer: 2

Dr. Gerald Lebovic, St. Michael's Hospital, Applied Health Research Centre

Thank you for the comments on our manuscript and for this suggestion.

In Supplementary Table 1 we now show the % missing in each of the analyses (in addition to the % missing for each variable). Three variables (waist-hip ratio, sedentary time spent per day and coronary artery calcium score) had >5% missing in the main analyses when also accounting for the missing data on education and place of birth. As suggested, we have performed sensitivity analyses using multiple imputation – the findings were largely similar to those of the main analyses.

Supplementary Table 1 Definitions and categorization of selected variables.

Variable	Categorization	n (%) missing^a	n (%) missing in main analyses^b
Sociodemographic information			
Age	Continuous in years	0 (0)	936 (3.1)
Sex	1. Women 2. Men	0 (0)	936 (3.1)
Place of birth	1. Not born in Sweden 2. Born in Sweden	811 (2.7)	936 (3.1)
Education	1. High school or less 2. Vocational education 3. University	845 (2.8)	
Cardiometabolic risk factors			
Diabetes status	1. Normoglycemia 2. Prediabetes (fasting glucose [6.1-6.9 mmol/L or glycated hemoglobin \geq 42 mmol/mol and <48 mmol/mol]) 3. Diabetes diagnosis by physician (self-reported in questionnaire) or glycated hemoglobin \geq 48 mmol/mol.	145 (0.5)	1064 (3.6)
Glycated hemoglobin	Continuous in mmol/mol	152 (0.5)	1070 (3.6)

Body mass index	Continuous in kg/m ²	2 (<0.5)	936 (3.1)
Weight status	1. Normal weight 2. Overweight 3. Obesity	2 (0)	936 (3.1)
Waist-hip ratio	Continuous	1567 (5.2)	2441 (8.1)
Systolic blood pressure	Continuous in mmHg	151 (0.5)	1022 (3.4)
Diastolic blood pressure	Continuous in mmHg	153 (0.5)	1024 (3.4)
Blood pressure level	Level as measured at inclusion in SCAPIS. 1. Normotensive (systolic blood pressure <140 mmHg and diastolic blood pressure <90 mmHg) 2. Grade 1 hypertension (Systolic blood pressure ≥140 mmHg and <160 mmHg or diastolic blood pressure ≥90 mmHg and <100 mmHg) 3. Grade 2 hypertension (systolic blood pressure ≥160 mmHg or diastolic blood pressure ≥100 mmHg).	153 (0.5)	1024 (3.4)
Current smoking	Self-reported in questionnaire. 1. No 2. Yes	954 (3.2)	1288 (4.3)
Time spent sedentary per day	Self-reported in questionnaire. Continuous (hours per day)	10441 (34.9)	10475 (35)
Coronary artery calcium score by computer tomography	Continuous	1197 (4.0)	1992 (6.6)
Total cholesterol	Continuous in mmol/L	82 (0.3)	1004 (3.4)
Low-density lipoprotein (LDL) cholesterol	Continuous in mmol/L	218 (0.7)	1134 (3.8)

HDL cholesterol	Continuous in mmol/L	85 (0.3)	1005 (3.4)
Creatinine	Continuous in mikromol/L	61 (0.2)	984 (3.3)

- a. n (%) missing values out of the total study population (n=29,955)
- b. n (%) missing in the main analyses adjusted for age, sex, place of birth and education.

Methods

“The proportion of missing data for the analyzed variables are shown in Supplementary Table 1 and ranged between 0 and 5% (0 to 8% in the main analyses using those with complete data on the variables assessed and adjusted for); the exception was time spent sedentary per day for which 34.9% of the participants had missing data. In post-hoc sensitivity analyses, we performed the main analyses with more than 5% missing data (those assessing waist-hip ratio, time spent sedentary per day and coronary artery calcium score) using multiple imputation (10 imputed datasets) created with chained equations. “

Results

“In the post-hoc sensitivity analyses using multiple imputation for logistic regression models with over 5% missing data, the results were similar to those in the main analyses: waist hip ratio (odds ratio per SD increase 1.54 [95% CI 1.20 to 1.98], time spent sedentary per day (odds ratio per hour increase 1.09 [95% CI 1.01 to 1.18] and coronary artery calcium score (odds ratio per 10 units increase 1.00 [95% CI 0.99 to 1.01])”

VERSION 3 – REVIEW

REVIEWER	Timoteo, Ana Teresa Ctr Hosp Lisboa Cent
REVIEW RETURNED	29-Jul-2021

GENERAL COMMENTS	I am satisfied with the answers and changes made in the manuscript.
---

REVIEWER	Lebovic, Gerald St. Michael's Hospital, Applied Health Research Centre, Li Ka Shing Knowledge Institute
REVIEW RETURNED	17-Aug-2021

GENERAL COMMENTS	Thank you for addressing my comments.
---------------------------------------